# Improving organic photovoltaic cells by forcing electrode work function well beyond onset of Ohmic transition

Chao Zhao [1,3], Cindy G. Tang[1], Zong-Long Seah[1], Qi-Mian Koh[2], Lay-Lay Chua [1,2], Rui-Qi Png [1] & Peter K. H. Ho [1✉]

As electrode work function rises or falls sufficiently, the organic semiconductor/electrode contact reaches Fermi-level pinning, and then, few tenths of an electron-volt later, Ohmic transition. For organic solar cells, the resultant flattening of open-circuit voltage ($V_{oc}$) and fill factor (FF) leads to a 'plateau' that maximizes power conversion efficiency (PCE). Here, we demonstrate this plateau in fact tilts slightly upwards. Thus, further driving of the electrode work function can continue to improve $V_{oc}$ and FF, albeit slowly. The first effect arises from the coercion of Fermi level up the semiconductor density-of-states in the case of 'soft' Fermi pinning, raising cell built-in potential. The second effect arises from the contact-induced enhancement of majority-carrier mobility. We exemplify these using PBDTTPD:PCBM solar cells, where PBDTTPD is a prototypal face-stacked semiconductor, and where work function of the hole collection layer is systematically 'tuned' from onset of Fermi-level pinning, through Ohmic transition, and well into the Ohmic regime.

[1] Department of Physics, National University of Singapore, Singapore, Singapore. [2] Department of Chemistry, National University of Singapore, Singapore, Singapore. [3] Present address: State Key Laboratory for Mechanical Behavior of Materials, Xi'an Jiaotong University, Xi'an, Shaanxi, People's Republic of China. ✉email: phyhop@nus.edu.sg

Recent advances in polymer donors and non-fullerene acceptors have produced organic photoactive layers (PAL) that give more than 15% PCE in single-junction solar cells[1–3]. Yet, the understanding and design of their charge collection electrodes have remained relatively primitive. New PALs are often tested with one of several favored electrode systems, for example, sol-gel ZnO as electron collection layer (ECL) and evaporated $MoO_3$ as hole collection layer (HCL) in inverted cells. But these electrode materials may not necessarily be best or most suited for manufacturing. A better understanding of contacts would yield new insights in disordered semiconductors, and reveal design rules to optimize performance of any selected PAL. This would ultimately enable development of better solution-processable electrodes that may be more suited to manufacturing.

In organic photovoltaic cells, electrodes set up a built-in potential ($V_{bi}$) that creates the internal electric field to generate photocarriers[4,5]. The $V_{bi}$ is determined by the difference of effective work functions ($\phi_{eff}$) between the electron ($e$) and hole ($h$) contacts, that is, $\phi_{eff,h} - \phi_{eff,e}$, where $\phi_{eff}$ is the energy difference between Fermi level (FL) of the specified electrode, and vacuum level (VL) of the semiconductor away from the contact[6]. $\phi_{eff}$ can differ significantly from the vacuum work function $\phi$ of that electrode, if interfacial dipole or charge transfer occurs between the electrode and the semiconductor[7–10]. The latter case causes pinning of $\phi_{eff}$ due to counteracting electric field set up by the accumulated carriers. Generally, $V_{bi}$ can be written as[11,12]: $V_{bi} = V_o - \Delta V_{el}$, where $V_o$ is the so-called "bare potential" term given by the difference between FL and VL at each of the two contacts, and $\Delta V_{el}$ is the total electrostatic band bending into the semiconductor due to carrier diffusion.

**Hole collection/injection polymers:**

| TAF | $R_1$ | $R_2$ | $X^-$ | $\phi$ |
|---|---|---|---|---|
| TFOMe-CF₃SIS | OMe | H | –SO₂N⁻SO₂CF₃ | 5.2 |
| TFOMe-C₂F₅SIS | OMe | H | –SO₂N⁻SO₂CF₂CF₃ | 5.4 |
| TFB-CF₃SIS | sec-Bu | H | –SO₂N⁻SO₂CF₃ | 5.35 |
| TFB-C₂F₅SIS | sec-Bu | H | –SO₂N⁻SO₂CF₂CF₃ | 5.55 |
| mTFF-C₂F₅SIS | H | CF₃ | –SO₂N⁻SO₂CF₂CF₃ | 5.75 |
| pTFF-C₂F₅SIS | CF₃ | H | –SO₂N⁻SO₂CF₂CF₃ | 5.9 |

**Photoactive materials:**

PBDTTPD          PCBM

**Fig. 1 Chemical structure of photoactive layer and TAF materials.** The value of $x$ denotes doping level of the triarylamine–fluorene (TAF) polymers, in hole per repeat unit: $x$ can vary between 0 (undoped) and 1 (fully doped). The value of $x$ is 0.7–0.8 for all TAF polymers studied here. $\phi$ is vacuum work function given in eV (±0.05 eV), measured by ultraviolet photoemission spectroscopy.

We have recently shown for P3HT:PCBM cells using HCLs with systematically "tuned" $\phi$ that both $V_{bi}$ and $V_{oc}$ track the $\phi_{eff}$ of the hole contact[13]. This indicates the hole contact is non-selective. If it were selective, that is, allowing only the correct carrier sign to exit, the majority-carrier density at the contact will rise with illumination intensity, decoupling $V_{oc}$ from $V_{bi}$[14,15]. Furthermore, we found that charge collection is subjected to an interfacial contact resistance that varies strongly with $\phi$[13]. A sharp Ohmic transition occurs a few tenths of an eV beyond FL pinning. The collection (or injection) resistance drops rapidly below the bulk resistance. Thus, while $V_{oc}$ rises strongly with $\phi$ and levels off at the onset of FL pinning, FF does so at the onset of the Ohmic regime. Together, they define an extended $\phi$ plateau where PCE broadly maximizes.

In this report, we have performed further detailed measurements in the Ohmic regime, and found that both the $V_{oc} - \phi$ and FF $- \phi$ plateaus can in fact tilt slightly upwards. Consequently, PCE can continue to improve, albeit slowly, with overdriving of $\phi$ in the Ohmic regime. The increase in $V_{oc}$ with $\phi$ arises as a consequence of soft FL pinning. This occurs in face-stacked polymer semiconductors, where their $\pi$-conjugation plane is parallel to the film plane, as distinct from edge-stacked semiconductors, where they are perpendicular. The $\pi$ orbitals of the frontier monolayer (ML) are exposed. This broadens the density of states (DOS), allowing coercion of FL up the DOS edge, and, consequently, gradual transition of the interface parameter[7–9], given by $S = \frac{d\phi_{eff}}{d\phi}$, from unity to 0. Thus, $\phi_{eff}$, $V_{bi}$, and $V_{oc}$ all continue to rise with $\phi$ well beyond the nominal FL-pinning threshold. On the other hand, the increase in FF with $\phi$ arises as a consequence of majority-carrier accumulation near the contact, as FL rises up the DOS edge. This causes carrier mobility to rise significantly, even in quality semiconductors with narrow widths, improving charge transport and collection. The phenomena together illuminate the importance of both interface and bulk DOS shapes for device physics, and opportunities for optimization beyond the Ohmic transition.

## Results and discussion

**Selection of model PAL system.** We employ PBDTTPD (chemical structures, Fig. 1) as model of a face-stacked polymer donor, and PCBM as model of an isotropic molecular acceptor. PBDTTPD is donor–acceptor (DA) polymer of the benzo[1,2-b:4,5-b']dithiophene (BDT) family that has yielded record organic solar cell efficiencies[16,17]. It comprises an electron-rich BDT donor moiety conjugated to an electron-deficient thiophene acceptor moiety. DA polymers generally exhibit a much weaker electronic coupling than homopolymers to conformational disorder, which results in better electronic transport[18,19]. However, they also exhibit a less congruous $\pi$-stacking owing to the DA dissymmetry[20–23]. This induces face-stacking in thin films, which aids charge transport through the film thickness direction[24]. PCBM provides a highly reliable photoinduced electron acceptor for PBDTTPD[25–27]. It gives rise to a robust donor–acceptor morphology, and temperature-independent photogeneration efficiency, which greatly simplify device analysis.

As polymer donor, PBDTTPD provides a contrast to P3HT in its polymer backbone orientation. Both their gas-phase HOMO energies are practically identical in the long-chain limit, *ca.* 6.15 eV, according to DFT/CAM-B3LYP/6-311G calculations (Supplementary Fig. 1). But their solid-state ionization energies ($I_E$) are considerably different. Ultraviolet photoemission spectroscopy (UPS) gives 5.3 eV for the PBDTTPD film, which is 0.65 eV larger than P3HT (Fig. 2). These values are unaltered by blending with PCBM. Thus, the difference may be attributed to a stable orientation effect, edge-stacking for P3HT and face-stacking for

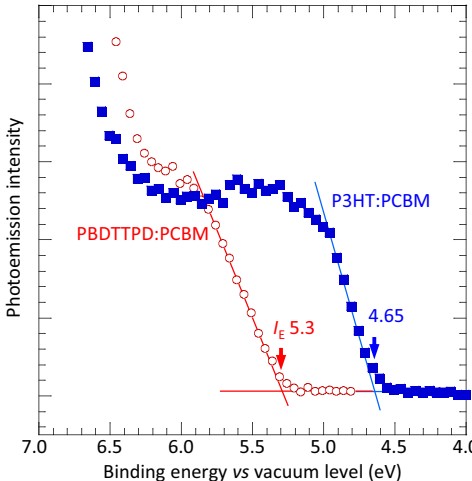

**Fig. 2 Ultraviolet photoemission spectroscopy of valence band.** Legend: red, PBDTTPD:PCBM (1:1.5 w/w) film; blue, P3HT:PCBM (1:0.8 w/w) film. Ionization energy ($I_E$) is marked by standard extrapolation of inflection point. Films were measured consecutively in identical configuration with same He-I radiation intensity.

PBDTTPD[28]. This is supported by the exclusion of PCBM from the surface layers of P3HT:PCBM films[29,30], but not PBDTTPD:PCBM films (Supplementary Fig. 2). Surprisingly, despite its weaker energetic dispersion, both PBDTTPD and PDBTTPD:PCBM films show a broader valence band edge than the P3HT counterparts. Hemi–Gaussian fitting gives the standard deviation width for the former to be 0.25 eV, and the latter 0.12 eV. However, the interpretation of these widths is not straightforward, because of packing effects on ionization energies[10,31].

**Estimation of transport DOS width.** To estimate the width of the relevant transport DOS, we measured the current-density–voltage ($JV$) characteristics of the films in the sandwiched diode configuration. If suitably strong hole injection layers (HIL) are used, the hole-only $JV$ characteristics of PBDTTPD:PCBM are well-behaved, similar to P3HT:PCBM. These follow the ideal Mott–Gurney law over one-and-a-half decades of $J$ up to 2000 mA cm$^{-2}$, that is, $J \sim \mu_{eff} (V - V_*)^2$, where $V_*$ is the apparent $V_{bi}$, and $\mu_{eff}$ is the effective carrier mobility (Supplementary Fig. 3). For PBDTTPD:PCBM, hole $\mu_{eff}$ is *ca.* $5 \times 10^{-4}$ cm$^2$ V$^{-1}$ s$^{-1}$, about three to five times as large as in P3HT:PCBM. The Gaussian disorder model thus suggests $\sigma/k_B T \lesssim 3$, i.e., $\sigma \lesssim 75$ meV, where $\sigma$ is the effective width of the transport DOS, that is, standard deviation of its forward edge (see discussions below). Good behavior suggests this edge extends at least $3.5\sigma$ below its effective center. Photothermal deflection spectroscopy indeed finds the optical DOS edge of "clean" polymer semiconductors is Gaussian out to $4\sigma$, but charge transfer and polaron excitations can intervene in the presence of acceptors[32–35]. Such excitations can be readily seen in PBDTTPD:PCBM by index-matched optical spectroscopy (Supplementary Fig. 4).

Thus, both PBDTTPD and P3HT films and their PCBM composites exhibit rather similar (and narrow) transport $\sigma$, but apparently widely differing surface $\sigma$. This may be rationalized by DOS broadening of the frontier ML due to variations at its interface, including from Coulomb fluctuations of fixed charges and free carriers in the adjacent doped polymer layer[36]. The frontier DOS dominates energy-level alignment at the contact, while the bulk DOS determines quasi-FL and carrier mobility inside the film. However, their differentiation is often neglected in the literature.

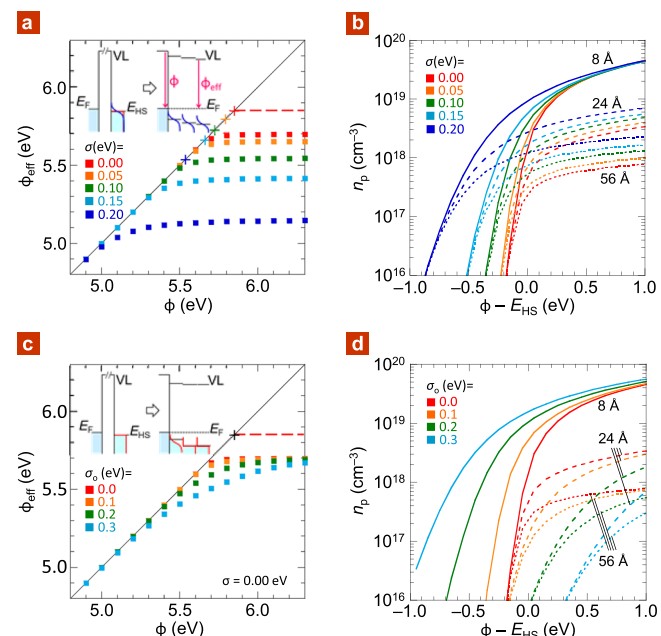

**Fig. 3 Computational study of DOS effects in Heaviside model. a** Fermi-level pinning plot for semiconductor with Heaviside DOS (5.85 eV; $5.5 \times 10^{20}$ eV$^{-1}$ cm$^{-3}$) convoluted with variable Gaussian width $\sigma$: Effective work function against vacuum work function of electrode. Red cross marks Heaviside energy $E_{HS}$. Other crosses mark valence-band-edge energy $E_V$ set up by corresponding $\sigma$. **b** Plot of hole density against $\phi$ offset by $E_{HS}$, for the cases in (**a**), at $z = 8$ Å (i.e., middle of ML0), 24 Å (ML1), and 56 Å (ML3). **c** Fermi-level pinning plot for Heaviside DOS where only ML0 is convoluted with variable Gaussian width $\sigma_o$. **d** Plot of hole density against $\phi$ offset by $E_{HS}$, for the cases in (**c**). $\phi_{eff}$ is evaluated at $z = 15$ nm. Other parameters: semiconductor dielectric constant, perpendicular to contact, $\varepsilon_r = 2.1$; charge-screening distance into electrode, $d_o = 5$ Å; ML thickness, $d_{ML} = 16$ Å; temperature, $T = 298$ K; numerical convergence quality, 1 mV.

**Computational study of DOS effects: Heaviside model.** To investigate the effects of these two DOS on energy-level alignment, we adopted the usual single-junction electrostatic model[37–39], but added surface broadening, allowing surface $\sigma_o$ to differ from bulk $\sigma$. We treat the polymer donor in the usual way as a series of MLs in thermal equilibrium with the electrode, taking into account image–charge polarization[40,41] and electrostatic band bending[11,42,43], but neglecting secondary influences from the opposite contact. The carrier density at thermal equilibrium is computed for each ML.

First, let us consider holes in a semiconductor with a hypothetical Heaviside DOS. This DOS rises abruptly from 0 to $5.5 \times 10^{20}$ eV$^{-1}$ cm$^{-3}$ at the Heaviside energy $E_{HS}$, set at 5.85 eV. It contacts an electrode with vacuum work function $\phi$, which sets up a local work function $\phi_{loc}$. This is the difference between the local FL and VL, which varies with distance ($z$) from the junction into the semiconductor. Under flatband condition for typical diode thicknesses, the electrostatic band bending vanishes at $z \approx 15$ nm. The value of $\phi_{loc}$ at this $z$ is thus taken to be the $\phi_{eff}$ that determines $V_{bi}$.

Figure 3a shows the plot of $\phi_{eff}$ vs $\phi$ for a temperature $T$ of 298 K. Holes are stabilized by the usual image–charge interaction energy: $\Delta E_{pol} = -\frac{1}{2}\frac{e^2}{4\pi\varepsilon_o\varepsilon_r}\frac{1}{2d}$, where the symbols have their usual meanings, which amounts to *ca.* 0.1 eV for the first ML ML0. Thus, most of the carriers in ML0 are trapped; only a small fraction is mobile, as noted previously[11,12]. For the Heaviside DOS, where $\sigma = 0$, the work function onset for pinning

($\phi_{pin}$) occurs at 5.7 eV. This occurs at the intersection of the $S = 1$ segment (Schottky–Mott limit) and the $S = 0$ segment (pinned-FL limit). The pinning depth given by: $\Delta E_{pin} = E_{HS} - \phi_{pin}$, is thus 0.15 eV at 298 K, which decreases to 0 as $T \to 0$ K.

When the Heaviside DOS is convoluted with $\sigma$, its frontier edge rises less steeply. Extrapolation of the inflection point gives an onset, which defines the valence-band-edge energy $E_V$, marked by crosses. The pinning depth now given by: $\Delta E_{pin} = E_V - \phi_{pin}$, becomes deeper, increasing from 0.15 to 0.27 eV as $\sigma$ increases from 0.0 to 0.15 eV. The $\Delta E_{pin}$ range is consistent with the energy-level alignment picture inferred from $V_{bi}$ measurements[44,45]. Concomitantly, the transition from the Schottky–Mott to the pinned-FL limit also becomes softer. Nevertheless, the value of $S$ over the interval $(\phi - \phi_{pin}) \in [0.2, 1.0 \text{ eV}]$ remains smaller than 20 meV per eV. Thus, the FL pinning is still fairly stiff, which can be represented by a fixed level outside the HOMO or LUMO band edge, as widely assumed[46]. Figure 3b shows the computed plot of accumulated hole density $n_p$ against $(\phi - E_{HS})$ at different depths into the semiconductor. These confirm that $n_p$ builds up primarily in ML0, with diffuse tail extending into higher MLs. As $\phi$ increases beyond $\phi_{pin}$, $n_p$ builds up quickly to a few $10^{19}$ cm$^{-3}$, proportional to $(\phi - \phi_{pin})$. This contact-induced carrier density, called "$\delta$-doping", can be directly observed by sub-gap electroabsorption spectroscopy[44,45].

Now, consider the same Heaviside semiconductor but with only ML0 subjected to Gaussian broadening of $\sigma_o$. Figure 3c shows the resultant FL pinning plot. Surface broadening is even more effective than bulk broadening to induce soft FL pinning. This is because of charge accumulation into ML0. Now $\phi$ has to be driven beyond $(\phi_{pin} + 2\sigma_o)$ to reach the final $\phi_{eff}$. Thus, the use of an electrode with extreme work function can advantageously coerce FL up the DOS edge in these cases to produce a larger $V_{bi}$. This phenomenon is inherent to disordered semiconductors. It does not require the presence of exogenous gap states, different from inorganic semiconductor surfaces[47]. Figure 3d shows that the hole density is largely confined to ML0. The $n_p$ inside the semiconductor diminishes (cf. Fig. 3b), because of screening by this hole density in ML0. This would eventually frustrate the formation of an Ohmic contact when $\sigma_o$ becomes too large. In summary, this study suggests that soft FL pinning can occur with practical consequences for $\sigma_o$ of a few tenths of an eV.

**Computational study of DOS effects: realistic semiempirical models.** To develop a simple yet realistic DOS model to parametrize $\sigma$ and $\sigma_o$, we treated the polymer donor as a distribution of conjugated segments, in the usual way, of length $\ell$ (in repeat units), each with own HOMO energy $E_{HOMO}$. For simplicity, we assume this distribution is uniform over the interval $[\frac{2}{3}<\ell>, 1\frac{1}{3}<\ell>]$, where $<\ell>$ is the mean conjugation length. This assumption imposes $\ell_{max}/\ell_{min} = 2$, ensuring each segment hosts only one transport site. We further assume that $<\ell>$ is given where: $\frac{dE_{HOMO}}{d\ell}\big|_{\ell=<\ell>} = k_B T$, and the hole-transport sites occupy 65 vol% of the donor matrix, which, in turn, occupies 50 vol% of the PAL. We then normalize this $\ell$ population with molecular volume to find states per unit volume, distribute in energy space, and convolute the native DOS with $\sigma$ to get the transport DOS model. Figure 4a shows the results for P3HT:PCBM and PBDTTPD:PCBM. Their DOS shapes differ, because of different energy dispersion in $\ell$. But their densities of hopping sites $N_{site}$ turn out to be fairly similar, *ca.* $2 \times 10^{20}$ cm$^{-3}$; and their DOS peak values also turn out to be similar, *ca.* 4 and $5 \times 10^{20}$ eV$^{-1}$ cm$^{-3}$, for P3HT:PCBM and PBDTTPD:PCBM, respectively.

To get the model for the surface ML, we convolute the native DOS with $\sigma_o$ instead of $\sigma$. We take $\sigma_o$ from an empirical equation:

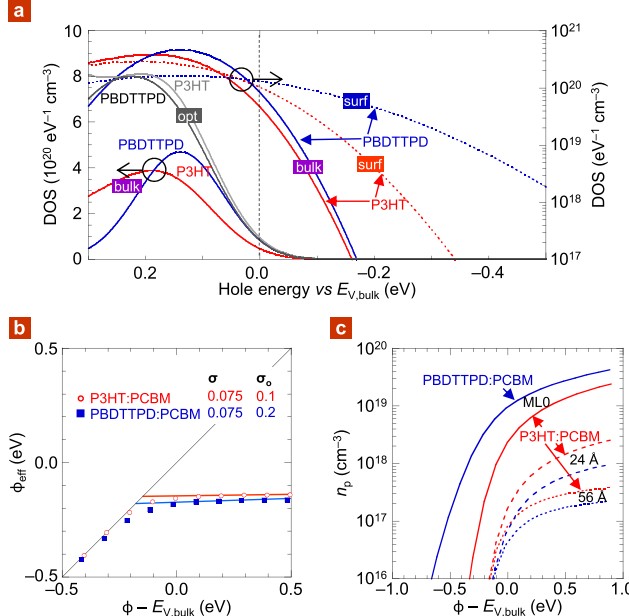

**Fig. 4 Computational study of DOS effects with semiempirical model. a** DOS models for P3HT:PCBM (red line) and PBDTTPD:PCBM (blue), both at 50 vol% donor: solid, transport DOS; dotted, first-monolayer DOS. Horizontal zero marks $E_V$ of the bulk semiconductor, $E_{V,bulk}$. The optical DOS model is also shown for comparison, arbitrarily located: P3HT:PCBM (gray line) and PBDTTPD:PCBM (black), normalized to same intensity. **b** Fermi-level pinning plots for the two sets of DOS models. **c** Plot of hole density against $\phi$ offset by $E_{V,bulk}$. $\sigma = 75$ meV, $\sigma_o = 100$ meV (P3HT: PCBM), 200 meV (PBDTTPD:PCBM). Other parameters are same as in Fig. 3, except for $d_o = 12$ Å; $d_{ML} = 16$ Å (P3HT, edge-stacked), 8 Å (PBDTTPD, face-stacked).

$\sigma_o = A \times \exp(-z/z_o)$, where $z$ is distance from junction to center of ML, $z_o$ is characteristic distance (5.8 Å), and $A$ is disorder amplitude (0.40 eV). This equation was parametrized by fitting the computed Coulomb fluctuation with distance from a polyelectrolyte layer (Fig. 4b in ref. [36]). It gives $\sigma_o$ to be 0.1 and 0.2 eV for P3HT:PCBM and PBDTTPD:PCBM, respectively, remarkably similar to the UPS results. In essence, the PBDTTPD surface layer is exposed to stronger multipolar Coulomb fluctuations because of its face-stacking. Our energy-level-alignment model then gives $S \approx 10$ and 50 meV eV$^{-1}$ for P3HT:PCBM and PBDTTPD:PCBM, respectively, as shown in Fig. 4b.

Thus, semiempirical DOS modeling predicts FL pinning to be softer for PBDTTPD:PCBM than P3HT:PCBM, due to energetic broadening of its frontier ML. Figure 4c shows the hole density in ML0 is consequently larger for PBDTTPD:PCBM than P3HT: PCBM, but of the same order of a few $10^{19}$ cm$^{-3}$ well beyond $\phi_{pin}$. The hole density inside PBDTTPD:PCBM is smaller than inside P3HT:PCBM, but of the same order of $10^{18}$ cm$^{-3}$ at $z = 2$ nm, which is sufficient for an Ohmic contact[44,45].

In summary, this study suggests that PBDTTPD:PCBM provides a contrast to P3HT:PCBM in its surface DOS due to polymer backbone orientation. Although soft pinning can be discerned in computed FL pinning plots for $\sigma \gtrsim 0.2$ eV[7–10], its possible relevance is disregarded, because the disorder threshold is so much larger than the typical transport $\sigma$ of 0.12 eV or smaller[48,49]. Here, we point out, however, that the typical energetic broadening of the frontier ML, particularly in face-stacked semiconductors—a common motif of "modern" DA polymers—potentially makes this phenomenon relevant.

**Fabrication and characterization of PBDTTPD:PCBM solar cells.** For experimental validation, we fabricated PBDTTPD:PCBM solar cells with a series of spin-on 20-nm-thick hole-doped triarylamine–fluorene (TAF) polymer[50] layers with different $\phi$ as HCL. The chemical structures of this family are shown in Fig. 1. A spin-on 100-nm-thick PBDTTPD:PCBM (1:1.5 weight/weight ratio) is used as PAL. An evaporated 30-nm-thick Ca, capped with 120-nm-thick Al, is used as ECL (see "Methods"). The TAF polymers are used in the self-compensated, heavily hole-doped form[51]. Tethered $CF_3SIS$ or $C_2F_5SIS$ counter-anions are used to confer improved ambient stability and control $\phi$. This family spans $\phi \in [5.2, 5.9\,eV]$ through a combination of semiconductor core and ion effects[52,53]. The lowest member, TFOMe-$CF_3SIS$, gives 5.2 eV, same as the reference poly(3,4-ethylenedioxythiophene):poly(styrenesulfonic acid) (PEDT:PSSH), and the highest member, pTFF-$C_2F_5SIS$, 5.9 eV. Because of the amorphous polymer backbone and identical doping level of 0.7–0.8 hole per repeat unit, i.e., $6 \times 10^{20}\,cm^{-3}$, film resistance and Schottky depletion width do not significantly change across the series. We measured the solar cell characteristics under AM1.5 irradiance of 100 mW cm$^{-2}$. Typical cell characteristics are shown in Fig. 5a, cell parameters in Fig. 5b, and values in Supplementary Table 1.

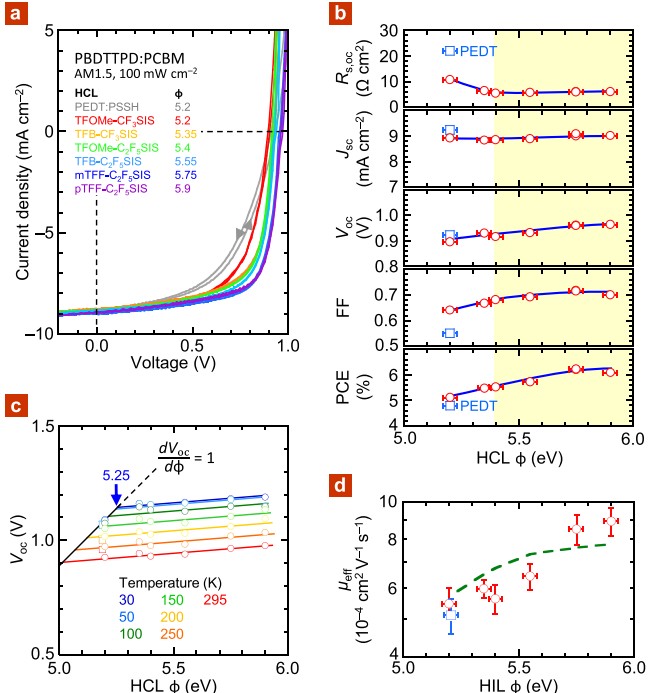

**Fig. 5 Characteristics of PBDTTPD:PCBM solar cells and hole-only diodes. a** $JV$ curves for different hole collection layers in cell configuration: ITO/20-nm HCL/100-nm PBDTTPD:PCBM (1:1.5 w/w)/30-nm Ca/Al, measured under simulated AM1.5 G irradiance of 100 mW cm$^{-2}$, spectral mismatch corrected, 298 K. **b** Cell parameters plotted against work function of hole collection layer. Data give population mean; standard error is smaller than symbol size. Blue lines are guides to the eye for TAF series. Yellow region corresponds to Ohmic regime. **c** $V_{oc}(\phi, T)$ plot measured at 1.0 sun. $V_{oc}$ data are averaged for forward and reverse sweeps. Typical uncertainty: ±0.01 V at high $\phi$; ±0.05 V at low $\phi$. Colored lines are fits with constant slope of 80 mV eV$^{-1}$. The black line ($\frac{dV_{oc}}{d\phi} = 1$) is anchored by low-temperature data set for $\phi = 5.2\,eV$. Below 50 K, $V_{oc}$ approaches $V_o$, which is the low-temperature $V_{bi}$. **d** Plot of effective hole mobility against work function of hole injection layer in hole-only diodes: ITO/20-nm HIL/100−120-nm PBDTTPD:PCBM (1:1.5 w/w)/Ag, for TAF series (circles) and PEDT:PSSH (square) as HIL. Error bar corresponds to standard error of mean. Green dashed line is from Gaussian disorder theory (see text).

The results show cell performance improves systematically with work function of the HCL as it increases well into the Ohmic regime. For reference, PEDT:PSSH gives $V_{oc}$ of 0.92 V, FF of 0.55, $J_{sc}$ of 9.25 mA cm$^{-2}$, and thus PCE of 4.8%, similar to, or better than, reported literature results for similar PBDTTPD molecular weights and processing conditions[16,17,54–56]. The morphology of the PAL can be further improved by hot spinning and/or solvent additives to reach an even higher $J_{sc}$ of 11 mA cm$^{-2}$[57–60]. However, this leads to a significant variability presumably due to donor–acceptor morphology drift that degrades our experiment. Thus, we use spin-cast films at room temperature without any solvent modifier. The open-circuit series resistance, given by $R_{s,oc} = \frac{dV}{dJ}|_{V=V_{oc}}$, of the cell with PEDT:PSSH is 22 Ω cm$^2$. This gives the sum of bulk charge-transport and interfacial charge-extraction resistances[13].

When PEDT:PSSH is substituted by TFOMe-$CF_3SIS$ having the same $\phi$, FF increases to 0.64, and the $JV$ hysteresis disappears. This improvement is associated with halving of $R_{s,oc}$ to 11 Ω cm$^2$, which we attribute to better hole collection by the TAF polymer without the tunneling layer on PEDT:PSSH[61,62]. As $\phi$ increases along the TAF series toward 5.9 eV, $R_{s,oc}$ decreases and then levels off at 5.5–6 Ω cm$^2$ for $\phi \gtrsim 5.35\,eV$. This marks the Ohmic transition of the PBDTTPD:PCBM contact[13]. However, $V_{oc}$ continues to rise steadily toward 0.96 V, and FF toward 0.72, while $J_{sc}$ remains constant at 9 mA cm$^{-2}$. Consequently, PCE increases from 5.1 to 6.1%. In contrast, the $V_{oc}$ of P3HT:PCBM cells declines slowly with $\phi$ beyond the Ohmic transition[13].

**Ruling out morphology and other artefacts.** To check for absence of any confounding variation in the PAL morphology, we spin-cast PBDTTPD:PCBM films of various thicknesses on fused silica, and on selected TAF films with $\phi$ of 5.35 or 5.75 eV, and measured our optical spectra (Supplementary Fig. 5). We found the $\pi \rightarrow \pi^*$ bandshape of PBDTTPD remains invariant, despite its strong dependence on film thickness. Spectral similarity is better than 1%, except for emergence of polaron band in the sub-gap of films on the HCLs due to interfacial hole doping. Thus, morphology artefacts can be ruled out.

To rule out device fabrication artefacts, we prepared "inverted" solar cells, where a self-compensated, electron-doped poly (fluorene–alt–benzothiadiazole) film ($\phi \approx 3.3\,eV$)[63] was first spin-cast as bottom ECL, then PBDTTPD:PCBM layer, then the TAF polymers top HCL was spin-cast from an orthogonal solvent, acetonitrile, over the PAL, followed by evaporation of Ag as top electrode (Supplementary Fig. 6). This eliminates any possibility that an HCL underlayer may have "primed" the PAL overlayer. Yet, we observed the same trends. As $\phi$ increases from 5.35 to 5.9 eV, $V_{oc}$ rises from 0.92 to 0.95 V, FF from 0.48 to 0.56, while $J_{sc}$ remains constant at 10.8 mA cm$^{-2}$. Thus, device fabrication artefacts can also be ruled out for the basic trends.

For comparison, solar cells with sol–gel ZnO as bottom ECL, and evaporated $MoO_3$ as top HCL, give $V_{oc}$ and FF of 0.93 and 0.60 V, respectively (Supplementary Fig. 7). Substituting $MoO_3$ with hole-doped mTFF-$C_2F_5SIS$ increases $V_{oc}$ to 0.97 V and FF to 0.66. Deep FL pinning is known to occur at the $MoO_3$/PAL interface[64]. Thus, these spin-on ultrahigh-work function polymers appear to have an inherent advantage over the evaporated ultrahigh-work function oxides.

**Evidence for soft Fermi-level pinning.** The $V_{bi}$ increases with $\phi$ of the HCL in the nominally pinned regime, as predicted by theory, confirming up-creep of FL at the hole contact. $V_{bi}$ was estimated by the method of "photocurrent inversion", which uses the saturated $V_{oc}$ at sufficiently low temperatures as proxy for $V_{bi}$[11,65]. This method applies for contacts that are nonselective

and non-injecting, as manifested by symmetry of the illuminated $JV$ characteristics about $V_{oc}$, that is: $J(V_{oc} + \delta V) = -J(V_{oc} - \delta V)$ for $\delta V \approx 0.1$ V, which occurs here for $T \lesssim 50$ K (Supplementary Fig. 8). The method fails if photogeneration efficiency falls strongly with decreasing temperature[66]. Figure 5c shows the $V_{oc}(\phi, T)$ plot, measured at 1.0 sun. The $V_{oc}$ values for $T \lesssim 50$ K converge to a limit. This limit corresponds to $V_o$, the low-temperature $V_{bi}$[11]. A break in the $\frac{dV_{oc}}{d\phi}$ slope for $T \lesssim 50$ K occurs at $\phi = 5.25$ eV, which gives the low-temperature $\phi_{pin}$. This separates a lower segment with $S = 1$ from an upper segment that has a small but non-zero value of $S$. In P3HT:PCBM cells, the upper $S$ value is 0[13]. Here, it is 80 mV eV$^{-1}$, as predicted by our DOS model. Clearly, the hole contact exhibits soft FL pinning for PBDTTPD:PCBM, but not P3HT:PCBM. At higher temperatures, $V_{bi}$ is subjected to $\Delta V_{el}$ loss, which scales logarithmically with carrier density at the contact, partially offsetting the FL upshift (Supplementary Fig. 9). This decreases $\phi_{pin}$ to *ca.* 5.1 eV at room temperature. In the absence of any FL upshift, the increasing $\Delta V_{el}$ loss would cause $V_{bi}$ to decline gradually with $\phi$, as observed in P3HT:PCBM cells[13]. Thus, softening of FL pinning enables the $V_{oc} - \phi$ plateau to tilt upwards.

**Evidence for hole-mobility enhancement.** However, drift–diffusion–generation modeling[14,38,67,68] with the above $V_o$ trend could not reproduce the FF results, unless we also allow the effective hole-mobility $\mu_{eff}$ to increase with $\phi$. To check directly for this supposed enhancement, we fabricated hole-only diodes with different PBDTTPD:PCBM film thicknesses, employing the TAF series as HIL, and evaporated Ag as hole-exit layer. We found that the $JV$ characteristics of PBDTTPD:PCBM films thicker than *ca.* 50 nm obey the Mott–Gurney equation with $V_*$ of $0.82 \pm 0.01$ V (Supplementary Fig. 10). Free fitting yields a Mott–Gurney index $m$ of $2.0 \pm 0.05$, where $m = \frac{d\log J}{d\log\left(V - V_*\right)}$ (Supplementary Fig. 11). Thus, the condition is satisfied[69] for reliable estimation of $\mu_{eff}$ from the Mott–Gurney equation: $J = \frac{9}{8}\varepsilon_r\varepsilon_o\mu_{eff}\frac{(V - V_*)^2}{d^3}$ (Supplementary Fig. 12). The results are plotted in Fig. 5d. They show that as $\phi$ increases from 5.2 to 5.9 eV, $\mu_{eff}$ increases from $6 \times 10^{-4}$ to $9 \times 10^{-4}$ cm$^2$ V$^{-1}$ s$^{-1}$. This indicates mobility enhancement induced by the contact.

**Origin of mobility enhancement.** We show that this effect is a manifestation of carrier mobility enhancement due to filling up of the DOS in disordered semiconductors. We considered the simple Gaussian disorder model, wherein carriers exhibit a constant mobility at low carrier density—the Boltzmann transport regime—which increases as the density approaches and exceeds a certain threshold that depends on $\sigma$[70–72]. For a Gaussian DOS with $\sigma/k_BT$ of 3, this threshold occurs at a reduced carrier density ($n_p/N_{site}$) of *ca.* $5.6 \times 10^{-3}$[73,74].

To model the expected size of effect, we first fitted the numerical results of Fig. 1 of ref. [73] to an empirical compact form: $\mu_p = \mu_{o,p}\ (1 + \alpha\ (n_p/N_{site})^\beta)$, where $\mu_{o,p}$ is the limiting hole mobility at low density, $\alpha$ is the hole-density coefficient, and $\beta$ is the exponent. This form describes the numerical results quite well. For $\sigma = 75$ meV, i.e., $\sigma/k_BT = 3$, we obtain $\alpha = 20$, $\beta = 0.50$, and $\mu_{o,p} = 4.3 \times 10^{-4}$ cm$^2$ V$^{-1}$ s$^{-1}$. The function is shown in Fig. 6a. Mobility begins to depart from the low-density limit at *ca.* $10^{16}$ cm$^{-3}$. A larger $\sigma$ would shift the onset to lower density.

With this density-dependent mobility, we computed the $JV$ characteristics for the PBDTTPD:PCBM hole-only diodes. The results give a $\mu_{eff} - \phi$ trend that matches the experimental one fairly well (dashed line, Fig. 5d). This is satisfying for a simple model without any tuning parameter. Thus, when $\phi$ increases

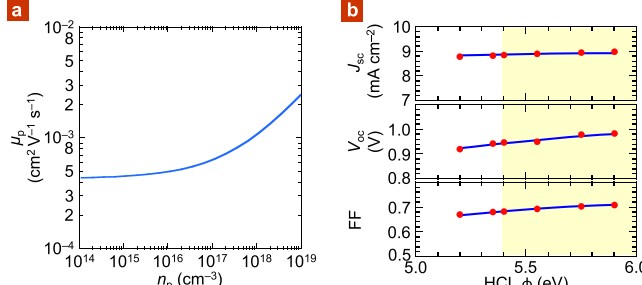

**Fig. 6 Simulation results. a** Hole-mobility simulation, from Gaussian disorder theory: $\mu_p = \mu_{o,p}\ (1 + \alpha\ (n_p/N_{site})^{0.50})$, where $n_p$ is hole density, $\mu_{o,p}$ is limiting hole mobility at low density, $\alpha$ is hole-density coefficient, and $N_{site}$ is transport site density. For parameter values, see caption of Table 1. **b** Drift–diffusion–generation simulation: red dots, simulation; blue line, experiment from Fig. 5b.

from 5.2 to 5.9 eV, the charge density in ML0 of PBDTTPD:PCBM increases from $7 \times 10^{18}$ to $3 \times 10^{19}$ cm$^{-3}$ (Fig. 4c). The corresponding mobile hole density ($N_p$), just beyond the contact, increases from $2 \times 10^{17}$ to $1.2 \times 10^{18}$ cm$^{-3}$, crossing the Boltzmann limit, enhancing carrier mobility over a substantial portion of the film.

To check for self-consistency with the solar cell characteristics, we also computed these using the same $\mu_p$ function. The other parameters, together with the apparent constant mobility needed to fit the characteristics $\mu_{p,app}$, are shown in Table 1. The values of $\mu_{p,app}$ are close to those of $\mu_{eff}$. The computed $V_{oc} - \phi$ and FF $- \phi$ trends also match the experimental ones very well, again without any tuning parameter (Fig. 6b). Thus, contact-induced mobility enhancement tilts the FF $- \phi$ plateau upwards in the Ohmic regime. This is the dominant factor improving FF of the solar cells in this regime.

The results here extend previous observations that both ultralow chemical doping[75,76] and high-current density[48,49] can enhance carrier mobility in diodes, as can high gate field in transistors[48,49]. Here, contacts with extreme work functions can accumulate carrier density sufficiently strongly to enhance carrier mobility, even in low-disorder semiconductors.

**Generalization.** To visualize carrier density profile within the PAL, we computed the depth-dependent $n_p$ and $n_n$ for various device conditions, and $\phi$ at both limits of 5.2 and 5.9 eV (Supplementary Fig. 13). We find for the hole-collection half of the cell that $n_p$ generally remains significantly larger for the larger $\phi$, under all conditions, which leads to the higher hole mobility. This contact-induced mobility enhancement should be general, independent of soft FL pinning. To check this, we fabricated P3HT:PCBM cells with different HCLs—PEDT:PSSH, TFOMe-C$_2$F$_5$SIS, and mTFF-C$_2$F$_5$SIS (Supplementary Fig. 14). The FF indeed systematically increases from $0.57 \pm 0.01$ for PEDT:PSSH to $0.70 \pm 0.01$ for mTFF-C$_2$F$_5$SIS, as $\phi$ increases from 5.2 to 5.75 eV. The final FF attained is unprecedented for these cells. As expected, $J_{sc}$ remains constant, while $V_{oc}$ declines marginally due to the $\Delta V_{el}$ loss.

In an attempt to push the mobility enhancement in PBDTTPD:PCBM films further, we fabricated hole-only diodes with even thinner PALs, down to 40 nm. However, the polymer morphology changes in these very thin films, as suggested by changes in their $\pi \to \pi^*$ spectra. This degrades $\mu_p$ severely, suppressing the expected enhancement (Supplementary Fig. 15).

Thus, simulation and experiment have firmly established two phenomena: (1) soft FL pinning, and (2) contact-induced mobility enhancement. Both these effects indicate that driving work function of the charge collection layer to extreme values

**Table 1 Input parameters for drift–diffusion–generation model.**

| HCL | $\phi$ (eV) | $V_o$ (V) | $N_p$ ($10^{17}$ cm$^{-3}$) | $\mu_{p,p}$ ($10^{-4}$ cm$^2$ V$^{-1}$ s$^{-1}$) | $\mu_{p,app}$ ($10^{-4}$ cm$^2$ V$^{-1}$ s$^{-1}$) |
|---|---|---|---|---|---|
| TFOMe-CF$_3$SIS | 5.2 | 1.13 | 2.1 | 7.2 | 5.2 |
| TFB-CF$_3$SIS | 5.35 | 1.17 | 3.5 | 8.0 | 5.6 |
| TFOMe-C$_2$F$_5$SIS | 5.4 | 1.18 | 4.2 | 8.4 | 5.7 |
| TFB-C$_2$F$_5$SIS | 5.55 | 1.20 | 7.2 | 9.7 | 6.1 |
| mTFF-C$_2$F$_5$SIS | 5.75 | 1.24 | 9.9 | 10.6 | 7.5 |
| pTFF-C$_2$F$_5$SIS | 5.9 | 1.25 | 12 | 11.2 | 7.8 |

$\phi$ is the measured vacuum work function of hole collection layer. $V_o$ is the measured "bare potential" component of $V_{bi}$ by electroabsorption spectroscopy. This is inferred to be 0.04 V larger at room temperature than at 50 K. $N_p$ is the mobile hole density at the hole contact computed from: $N_p = N_o (\phi - \phi_{pin})$, for $\phi > \phi_{pin}$, where $N_o = 1.5 \times 10^{18}$ cm$^{-3}$ eV$^{-1}$, and $\phi_{pin} = 5.1$ eV. $\mu_{p,p}$ is the hole mobility evaluated at the hole contact. General hole mobility is taken from: $\mu_p = \mu_{p,o} (1 + \alpha (n_p/N_{site})^\beta)$, where $\mu_{p,o} = 4.3 \times 10^{-4}$ cm$^2$ V$^{-1}$ s$^{-1}$, $\alpha = 20$, $\beta = 0.50$, and $N_{site} = 2.0 \times 10^{20}$ cm$^{-3}$, for $\sigma = 75$ meV (see text). Electron mobility is taken to be constant at $2.0 \times 10^{-4}$ cm$^2$ V$^{-1}$ s$^{-1}$. The mobile electron density at the electron contact $N_n$ is taken to be constant at $2.0 \times 10^{17}$ cm$^{-3}$. Bimolecular electron–hole recombination is assumed: geminate recombination rate constant, $10^5$ s$^{-1}$; bimolecular recombination rate constant, $2 \times 10^{-11}$ cm$^3$ s$^{-1}$. The solar cell characteristics can also be fitted with constant apparent hole-mobility values $\mu_{p,app}$, as shown in the last column.

well beyond the Ohmic transition can significantly improve $V_{bi}$ and $V_{oc}$ of organic photovoltaic cells. The strategy is simple, and can be used to generate best performance, even for low-molecular-weight polymer semiconductors. However, there must ultimately exist a trade-off from the increased minority-carrier recombination in the near-contact region, though this does not yet pose a problem here, presumably because bimolecular recombination is very weak[11,77].

## Methods

**Materials**. PBDTTPD (1-Material), P3HT (Rieke Metals), and PCBM (1-Material) were obtained from commercial sources, and used as received. PBDTTPD was measured to have $M_n = 29$ kD by gel permeation chromatography in 1,2,4-tri-chlorobenzene at 160 °C, and also chloroform at 40 °C. 1:6 wt/wt PEDT:PSSH solution (Clevios P VP Al 4083, Heraeus Precious Metals GmbH) was purified by dialysis against 1-M semiconductor-grade HCl solution, followed by Millipore® water, through a 12-kDa molecular-weight-cutoff membrane to remove cation and acid impurities[78]. TAF polymers were synthesized following procedures reported in Tang et al.[51]. To hole dope the TAF polyelectrolytes, the dry TAF polymers were baked on a hotplate at 120 °C in an N$_2$ glovebox for 1 h, then dissolved in anhydrous acetonitrile to give a 20 mM solution, mixed with 1.0 equivalent of nitrosonium hexafluoroantimonate (NOSbF$_6$), also dissolved in anhydrous acetonitrile (40 mM), then purified by precipitation in diethyl ether and redissolution in acetonitrile, twice, to give the respective self-compensated, hole-doped TAFs. The final acetonitrile solutions were used directly for spin casting. The doping levels were evaluated by UV–Vis spectrophotometry[51].

**Solar cell fabrication and measurements**. For PEDT:PSSH as HCL, 40-nm-thick films were spin-cast from the PEDT:PSSH solution onto oxygen-plasma-cleaned ITO substrates, and baked on hotplate at 140ºC in air for 10 min; for TAF as HCL, 20-nm-thick films were spin-cast from acetonitrile solutions onto the ITO substrates in the N$_2$ glovebox. Then a solution of PBDTTPD:PCBM (1:1.5 w/w; 30 mg mL$^{-1}$) was spin-cast over the HCLs in the glovebox to give 100-nm-thick films. A 30-nm-thick Ca layer, followed by 120-nm-thick Al layer, was thermally evaporated through a shadow mask at a pressure of $10^{-6}$ mbar to define eight 4.29-mm$^2$ pixels on each substrate. Ag paint was applied on the metal electrodes to eliminate external contact resistance. $JV$ characteristics of the cells were collected using a semiconductor parameter analyzer (Keithley 4200) in a N$_2$ chamber, in the dark, and under 100 mW cm$^{-2}$ simulated AM1.5 solar irradiance (Oriel Sol2A), spectral mismatch corrected (factor, 1.08). Some of the cells were characterized from room temperature (298 K) to 30 K in a cryostat.

**Hole-only diode fabrication and measurements**. The ITO/HCL/PBDTTPD:PCBM (1:1.5 w/w) stacks were built as described above, but with PBDTTPD:PCBM thicknesses varied over 45–100-nm range. These thicknesses were measured as the step thickness of the scratched stack profile; repeatability was better than ±5 nm. Then a 120-nm-thick Ag layer was thermally evaporated through a shadow mask at a pressure of $10^{-6}$ mbar to define eight 4.29-mm$^2$ pixels on each substrate. Ag paint was applied on the metal electrodes to eliminate external contact resistance. $JV$ characteristics of the diodes were collected on a probe station in the N$_2$ glovebox using a semiconductor parameter analyzer (Keithley 4200).

**Ultraviolet photoemission spectroscopy (UPS)**. UPS was performed on films in an ESCALAB UHV chamber equipped with an Omicron EA 125 U7 hemispherical electron energy analyzer at a base pressure of $10^{-9}$ mbar. 10-nm-thick PBDTTPD:PCBM and P3HT:PCBM films were spin-cast in N$_2$ glovebox, following the standard protocol, onto O$_2$-plasma-cleaned Au-coated Si substrates. The substrates were transported in N$_2$ and loaded into the UPS chamber without exposure to

ambient air. UPS was performed using He-I radiation (21.21 eV). The photo-emission normal to the film surface was collected and analyzed at a pass energy of 5 eV, giving a resolution of 50 meV.

**Low-temperature $JV$ measurements**. Low-temperature $JV$ measurements were performed in a closed-cycle helium cryostat (Janis APD HC-2). The cells were loaded into the cryostat in the N$_2$ glovebox, and evacuated to $10^{-6}$ mbar by a turbomolecular pump. $JV$ characteristics were measured at 1 sun following the standard protocol.

## Data availability

Source data for all figures are available from the corresponding author.

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

## Acknowledgements

We thank Qiu-Jing Seah for synthesizing the TAF materials. This research is partially supported by the National Research Foundation, Prime Minister's Office, Singapore under its Competitive Research Programme (CRP Award No. NRF-CRP 11-2012-03: R-144-000-339-281, R-143-000-608-281).

## Author contributions

C.Z. and Z.-L.S. fabricated and characterized the devices. C.Z. performed the device simulations. C.G.T. performed the DFT calculations. Q.-M.K. provided materials support. R.Q.P. led the device effort, L.-L.C. led the materials effort, P.K.H.H. led the theory effort. C.Z., R.Q.P., and P.K.H.H. wrote the paper. All authors discussed the experiments and results.

## Competing interests

The authors declare no competing interests.
