## [Peer Review File · Nature Communications]

Reviewers' comments:

Reviewer #1 (Remarks to the Author):

Comments to Authors

In the manuscript titled "Filling density-of-states at contacts for higher open-circuit voltage and fill factor in organic solar cells", the authors increased open-circuit voltage and fill factor of solar cell devices with high-performance donor polymers like PBDTTPD. The reviewer thinks that researchers in this field would be very interested in the contents of this manuscript. Therefore, the reviewer would strongly recommend the publication of this paper to Nature Communications after the following minor revision. These comments are not to criticize but to strengthen the manuscript.

1. In page 8, the authors mentioned that Jsc and FF can be further improved by morphology optimization. The reviewer recommends the authors to briefly explain the details of morphology optimization(i.e. the change of experimental conditions, etc.).

2. The author seems to miss a relevant paper on organic solar cell based on PBDTTPD. The reviewer recommends that the authors include the following references in "Introduction" to solidify the manuscript.

[1] Chem. Mater., 2014, 26, 2299-2306: Paper about the morphology of PBDTTPD-PCBM bulk-heterojunction solar cells

[2] Organic Electronics, 2017, 42, 42-46 : Paper about eco-friendly fabricated PBDTTPD:PC71BM solar cells

Reviewer #2 (Remarks to the Author):

The manuscript presents the impact of varying the contact work function of the hole-extracting contact on the solar-cell performance of PBDTTPD:PCBM organic solar cells. Claims are made that with increasing contact work function, the open-circuit increases beyond the Fermi-level pinning regime. Also the hole mobility increases, attributed to trap filling at the electrode interface, leading to an increased fill factor. As such, it is concluded that electrode work functions for hole contacts should substantially exceed the valence-band onset of the donor polymer.

I have several problems regarding the generality of these claims and the presented evidence. First of all, it appears that similar Voc and FF have been achieved with a conventional PEDT:PSSH electrode: W. R. Mateker et al., Energy Environ. Sci., 6, 2529–2537 (2013). This raises questions if such high work functions are really necessary.

More direct evidence of the Fermi level moving inside the DOS should be given by UPS measurements, with the horizontal scale w.r.t. the Fermi level. I would further like to note that UPS measurements are not suitable for quantifying the width of the DOS, since many factors can result in broadening of the signal.

The drift-diffusion simulations should be able to reproduce the effect of the contact Fermi level on Voc without adjusting the built-in potential. This would be far more convincing.

The authors argue that the mobility also enhances as a result of the contact work function, however, Fig. S5 shows that for thin layers there is no significant influence. Especially for thin layers, filling traps near the interface would have a substantial effect on the mobility. For thicker layers, the injecting interface becomes less important. The small observed mobility increase may be specific to this polymer:fullerene blend and its morphology influenced by the electrode surface.

Overall, I am not convinced that morphological difference can be ruled out as the origin of the observed changes in Voc and FF. In addition, the observed performance increase compared to literature (with PEDT electrode) does not appear sufficiently substantial to warrant publication of these results in Nature Communications.

Reviewer #4 (Remarks to the Author):

The authors employ a previously developed electrostatic model for semiconductor/electrode systems, to investigate the influence of band edge broadening on Fermi level (FL) pinning. As correctly noted by the authors, the FL pinning behavior of a Gaussian–Lorentzian DOS has been analyzed previously (reference 14 in the manuscript). Next, it is claimed that they characterize the stiffness of FL pinning and their consequences, which have been neglected thus far. From my knowledge of literature (which yet may be incomplete), I cannot see this claim supported and thus no conceptual novelty in this work beyond established ones. For instance, the impact of gap state density (or tailing of the majority semiconductor DOS into the gap) on the FL pinning behavior was indeed already described in ref. 14 of the present manuscript (Fig. 4b and 5c therein), and more recently discussed in *J. Phys. D: Appl. Phys.* 50, 2017, 423002. Additionally, the notion provided in the present manuscript has essentially already been put forward by some of the authors in *Nat. Commun.* 9, 2018, 3269 (compare text in that publication versus the present statement "The improvement is post Fermi level pinning and post-Ohmic transition.").

Yet, I do consider the present work an important experimental demonstration that the above expected correlations hold, to make the community working on organic solar cells aware of it. For this, I would recommend a different, more focused, journal for dissemination, such as *Advanced Materials Interfaces*, *ASC Applied Materials Interfaces*, or *Materials Today-Energy*. Yet, before the authors do submit their work elsewhere, I would strongly suggest that they provide direct experimental evidence for the different width of the DOS of the two material combinations studied, as presently this is only assumed from modeling. Otherwise, the conclusions reached by the authors is not strongly supported by data.

Reviewer #1 (Remarks to the Author):

We thank this reviewer for his/her positive comments and for pointing out two key paper that studied the morphology optimization of PBDTTPD:PCBM bulk-heterojunction, which we have now cited. We have also added some more discussion on morphology optimization.

In the manuscript titled "Filling density-of-states at contacts for higher open-circuit voltage and fill factor in organic solar cells", the authors increased open-circuit voltage and fill factor of solar cell devices with high-performance donor polymers like PBDTTPD. The reviewer thinks that researchers in this field would be very interested in the contents of this manuscript. Therefore, the reviewer would strongly recommend the publication of this paper to Nature Communications after the following minor revision. These comments are not to criticize but to strengthen the manuscript.

1. In page 8, the authors mentioned that J_{sc} and FF can be further improved by morphology optimization. The reviewer recommends the authors to briefly explain the details of morphology optimization (i.e. the change of experimental conditions, etc.).

Response: Fixed.

2. The author seems to miss a relevant paper on organic solar cell based on PBDTTPD. The reviewer recommends that the authors include the following references in "Introduction" to solidify the manuscript.

[1] Chem. Mater., 2014, 26, 2299-2306: Paper about the morphology of PBDTTPD-PCBM bulk-heterojunction solar cells

[2] Organic Electronics, 2017, 42, 42-46: Paper about eco-friendly fabricated PBDTTPD:PC71BM solar cells

Response: Fixed.

Reviewer #2 (Remarks to the Author):

We thank this review for his/her critical comments that have triggered improvement our manuscript. We have added inverted devices, and optical spectroscopy measurements to firmly conclude that the device performance improvement does indeed come from contact effect rather than morphology changes, and clarified the manuscript accordingly.

1. *I have several problems regarding the generality of these claims and the presented evidence. First of all, it appears that similar V_{oc} and FF have been achieved with a conventional PEDT:PSSH electrode: W. R. Mateker et al., Energy Environ. Sci., 6, 2529–2537 (2013). This raises questions if such high work functions are really necessary. Overall, I am not convinced that morphological difference can be ruled out as the origin of the observed changes in V_{oc} and FF*

Response: We thank the referee for the comments. Two issues that we need to address. One, is such high work function necessary? For conventional PEDT:PSSH electrode to achieve reported V_{oc} and FF, it was claimed that morphology optimization through a combination of hot spinning, solvent modifiers, and high Mn materials was necessary. This recipe is difficult to reproduce on large areas, and the morphology obtained appeared to be metastable. However, in our work, we can routinely exceed best performance in FF and Voc without any of these difficult conditions. This is an important step driving towards a new understanding of possibilities in device physics, and repeatable manufacturability.

Two, is it really a contact effect? Spurred by Referee's doubt, we have added more experiments: (i) inverted devices using the ultrahigh work-function layers as overlayer instead of underlayer (Supplementary figure 8), where it is not possible for the overlayer (made using orthogonal solvents) to systematically influence the morphology of the underlayer; and (ii) detailed optical spectroscopy of PAL with different layers and on substrates with different WF (Supplementary figure 5), which shows the different WF-substrates do not induce change in the observable π - π^* spectrum. Thus the observed improvement in FF and Voc is contact dominated. By filling-up DOS, which has never previously been demonstrated, we are able to achieve the high V_{oc} and FF without further morphology refinement. This helps liberate the potential of high-performance materials systems.

We have performed more detailed simulations (Fig 3). The new data and computations reinforced our earlier conclusions on the strong contact-induced effects. These have not previously been observed, because researchers did not have the systematically 'tunable' WF layers in the ultrahigh range that are now available.

2. *More direct evidence of the Fermi level moving inside the DOS should be given by UPS measurements, with the horizontal scale w.r.t. the Fermi level. I would further like to note that UPS measurements are not suitable for quantifying the width of the DOS.*

Response: We thank the referee again for the suggestion. The idea is nice. However, the challenges are twofold. The expected Fermi level shift inside the semiconductor DOS is only 50 meV, which is the threshold of reliable detection by UPS. Even more challenging, the photoactive layer generates a very high density of electron-hole pairs under He I radiation (maybe more than 10^{17} cm⁻³, 100-fold of sun), which appears to cause the Fermi level to drift back to the mid-gap. Hence, in the manuscript, we

employed built-in potential measurements (resolution better than 0.01 V) by photocurrent inversion, which allowed us to track the overall *shift* in energy-level alignment with WF of the hole contact with sufficient accuracy. This is a direct measurement, which joined to self-consistent drift-diffusion simulations, confirms soft Fermi-level pinning is primarily responsible for the improvement in cell performance.

UPS measurements provides a measure of the disorder width of the frontier DOS at the surface, from fitting to the leading edge which is relevant to Fermi level pinning. This is quite well accepted in the field (see for example, Fahlman and co-workers, *Adv Mater* 21 (2009) 1450; Kahn and co-workers, *Mater Sci Eng R* 64 (2009) 1 1450). The only assumptions required are the hole lifetime is not limiting (aka Heisenberg Uncertainty principle), and the photoionization cross section is constant across the leading edge. The actual width at a buried interface depends on the other material (polarization fluctuations, etc). Therefore we use these results only to guide the development of our model DOS, and the range of “realistic” disorder values that need to be tested in our model. We have rewritten this section more carefully now to distinguish between surface and bulk DOS, correcting an earlier mistake. We thank the referee for drawing our attention to this aspect. In case the Referee is thinking about transport, the UPS disorder DOS width does not give the transport width. It is a joint DOS summing over all wave functions, whereas the relevant transport DOS deals only with the lowest energy wavelength of the sites accessible to carriers inside the percolating path. And we are certainly not referring to the total band width of the DOS, which is indeed not possible to quantify from UPS. We have revised the manuscript to clarify these point.

Anyway, to clarify our own doubt whether the frontier shape is indeed Gaussian-like, we have checked that the Mott–Gurney law is obeyed over a wide range of hole current densities (10–1,000 mA cm²), and quantified the constraints on the Gaussian disorder width. We have also performed index-matched optical spectroscopy to check directly the joint DOS measurements by (Supplementary figure 4). We can confirm the Gaussian assumption is good to at least 4 standard deviations. This forms a basic tenet on which our work is built. Anyway, it is not surprising. The Gaussian model has been generally accepted in the field for “clean” materials (e.g. see the work of Bassler, Coehoorn, Blom, Baranovskii in the new references).

3. *The drift-diffusion simulations should be able to reproduce the effect of the contact Fermi level on V_{oc} without adjusting the built-in potential.*

Response: No. Unfortunately, this is not possible in the present implementation of the drift-diffusion model without combining with an injection model. The drift-diffusion model requires the equilibrium carrier densities to be specified at the interfaces as input parameters, not dependent parameters. A V_{bi} is thus needed (actually V_o , the “bare” potential term that is stripped of the electrostatic band bending contribution, following our method to implement this in a globally self-consistent manner) to tell the drift-diffusion the energy gap. This V_o is guided by V_{bi} measurements at low temperature where the electrostatic losses become small. It is NOT an ‘adjustable’ fitting parameter as the referee thought. By pinning down all key parameters by experiments, our implementation of the drift-diffusion-generation equations in essence

turns it into a zero-free-parameter model, whose results and insights are thus reliable. We have revised the manuscript to clarify this point.

4. *The authors argue that the mobility also enhances as a result of the contact work function, however, Fig. S5 shows that for thin layers there is no significant influence. Especially for thin layers, filling traps near the interface would have a substantial effect on the mobility. For thicker layers, the injecting interface becomes less important. The small observed mobility increase may be specific to this polymer:fullerene blend and its morphology influenced by the electrode surface.*

Response: No, the different WF surface does not alter the morphology of the photoactive layer. Index-matched transmission optical spectroscopy shows the π - π^* band shape of various thicknesses (45–170 nm) of PBDTTPD:PCBM films do not change (<1 %) with WF of the substrate film, except for the emergence of the polaron band in the gap due to hole-doping of the contact (Supplementary Fig. 5). This rules out the possibility that the ultrahigh-WF substrate induces changes in the morphology of the deposited PBDTTPD:PCBM overlayer. The “small” mobility improvements are in fact large enough to decrease the bulk resistance of the cells, improving the FF significantly. Spurred by referee’s doubt, we show all our data analysis now in Supplementary Figs 9-11. The uncertainty in mobility is less than 10%. The systematic evolution with work function over 60% is at first sight shocking, but it is real, and in fact consistent with the Gaussian disorder model. Enhancement of mobility with carrier density has been observed in organic FETs and single-carrier diodes, and now finally in organic solar cells despite the large photogenerated carrier densities at 1-sun. Therefore the effect is relevant to organic solar cell physics.

Reviewer #4 (Remarks to the Author):

We thank this review for his/her critical comments to improve our manuscript. We have now revised the manuscript accordingly to explain the post-Ohmic transition aspect better.

1. *For instance, the impact of gap state density (or tailing of the majority semiconductor DOS into the gap) on the FL pinning behavior was indeed already described in ref. 14 of the present manuscript (Fig. 4b and 5c therein), and more recently discussed in J. Phys. D: Appl. Phys. 50, 2017, 423002*

Response: What is different between our work, and the earlier beautiful work on which ours stand, is: (1) emphasis on the consequences and opportunities of soft pinning, particularly for organic solar cells, where every 0.01 V matters, which hasn't been picked up by other authors, (2) emphasis on carrier-density-induced mobility enhancement in organic solar cells, and also diodes, due to contact carrier accumulation, which has never been observed previously. To clarify these new pieces of physics, we have developed a novel semiempirical approach to model the DOS, a globally-self-consistent approach to do drift-diffusion-generation simulation. For the equilibrium energy-level alignment, although we employ the same electrostatic model, we have added a new feature of additional broadening into the frontier monolayer representing orientation effects in modern donor polymers. This crucial feature is missing from earlier work, and even in the recent J Phys D: Appl Phys paper cited by the referee. As a consequence, the importance of soft Fermi-level lay undiscovered for so long. We have performed further simulations now, see expanded Fig 3, to emphasize the consequence of soft Fermi-level pinning not only on alignment but also on carrier densities, both of which have not been previously explored. We thank the referee for this comment, which has prompted us to clarify our manuscript better.

2. *Additionally, the notion provided in the present manuscript has essentially already been put forward by some of the authors in Nat. Commun. 9, 2018, 3269 (compare text in that publication versus the present statement "The improvement is post Fermi level pinning and post-Ohmic transition.").*

Response: Not really. Previously, we learnt that we must reach Ohmic transition, beyond Fermi-level pinning, to get best performance for V_{oc} and FF. That result is general. Now we learnt something more surprising—for face-stacked polymers, we can exploit a post-Ohmic effect to squeeze even more mileage from the photoactive layer system using a very simple approach of extreme WF collection layers, whose physics we have also unravelled in the manuscript. The essential two new pieces of physics are: (i) soft Fermi level pinning, and (ii) contact carrier-density mobility enhancement. Because these face-stacked polymers are now the most important high performance materials, this result becomes very powerful for optimizing solar cell performance in a novel way that has not previously been unanticipated.

REVIEWER COMMENTS

Reviewer #2 (Remarks to the Author):

The authors have clarified some doubts I had with respect to the the original submission. In particular the authors have strengthened their argument that the improved solar-cell performance is indeed an effect induced by the contacts, and not by the morphology. Furthermore, they have clarified why they use V_0 as a parameter in their drift-diffusion simulations. While I still believe that it would be desirable to verify the experimentally observed effects with drift-diffusion simulations with a constant energy gap instead of a variable V_0 (such as the model put forward by Blakesley and Neher, ref. 41), using an experimentally determined V_0 would be acceptable in my opinion.

I am, however, still not overly convinced by the mobility increase. In Supplementary Fig. 12, the effect of the contact work function on thin films is not visible, while the mobility in thin films should be more sensitive to the charge density at the contact compared to thick films. This set of data would rather point to a transition from a slightly non-Ohmic regime to an Ohmic regime, as opposed to the "post-Ohmic" regime as the authors put forward. In SCLC measurements it is hard to distinguish whether an increased current is due to a higher injection rate or an increased mobility. Usually thickness dependence gives a good clue, but in this case the evidence would be in favor of an injection limitation due to slightly non-Ohmic contacts. This would of course also have an effect on the mobility due to the reduced charge density, but this would not be the dominant effect in the magnitude of the current.

Another implication of a density-dependent mobility would be a non-constant mobility over the measured current density range, as is assumed in the fitting.

These points of criticism are not to say that the results are invalid, and the authors have now convincingly demonstrated that for these face-stacked polymers extreme contact work functions can be used to extract more performance from such solar cells. I would, however, avoid using terms like "post-Ohmic" in a revised version of the manuscript.

Some additional questions:

- How would a more conventional MoO₃/Ag electrode compare in terms of performance in an inverted device architecture? MoO₃ has also been reported to have an extremely high work function.
- What assumptions/values are used for the bimolecular recombination rate in the simulations? I could not find these details (maybe I have overlooked), but this is important information.

Reviewer #4 (Remarks to the Author):

Without question, the authors have improved their manuscript. However, I feel very uncomfortable by the way the authors push the "novelty" aspects of their work with respect to prior art, and I still miss a proper discussion of some established knowledge.

But rather than contemplating my feelings, let's summarize the facts:

The authors write: "... we demonstrate that for contacts that exhibit 'soft' FL pinning, driving ϕ well beyond the Ohmic transition can further enhance both V_{bi} and V_{oc} by advantageously forcing FL up the DOS tail."

Comment: In the entire introduction the authors fail to clearly explain state-of-the-art knowledge of FL pinning. While in the "old" literature FL pinning was often alleged to be abrupt (seen from the z-like curves plotted with an abrupt change in slope), it was later on clearly stated (also in papers cited by the authors) that any density of states that is not abrupt leads to a further movement of

FL with electrode work function change. This is irrespective of shape of the majority DOS (e.g., Gaussian), and particularly governed by sample-dependent gap state density (which can be intrinsic or extrinsic, see work by the Ueno group), its shape and actual density. This, for instance, was shown explicitly for an inorganic semiconductor quite recently (doi.org/10.1103/PhysRevMaterials.3.074601; and I just realize that even the term "soft FL pinning" was used there).

So there is no new physics here, but the authors demonstrate the impact of "soft" FL pinning on V_{bi} and V_{oc} of OPV. This has not been shown yet.

The authors write: "Furthermore, we demonstrate that in this post-Ohmic transition regime, the apparent mobility of the majority carrier can also be forced to increase."

Comment: It is established by numerous publications that (low) doping of an organic semiconductor increases carrier mobility due to filling of shallow trap states. This is, from a fundamental point of view, generally valid and fully independent of how doping is done, whether it is with dopants or by inducing FL pinning (the present case).

So there is no new physics here, but the authors demonstrate the impact of this contact-doping on the performance of OPV.

In conclusion, the present submission is good at explaining known physics for non-experts in electronic properties and working in OPV engineering. Whether this suffices for publication in *Nat. Commun.*, I don't know. In any case, before publication in any journal, I strongly recommend that the authors provide a suitably improved representation of prior art (and including the relevant references, which are presently missing with respect to FL pinning and gap states, as well as doping-induced carrier mobility changes) along the lines of my comments above.

Response to review comments

Reviewer #2 (Remarks to the Author):

We thank this review for his/her incisive comments to improve the science in our manuscript. We have performed these additional experiments and calculations now, and incorporated into the revised manuscript.

1. *The authors have clarified some doubts I had with respect to the original submission. In particular the authors have strengthened their argument that the improved solar-cell performance is indeed an effect induced by the contacts, and not by the morphology. Furthermore, they have clarified why they use V_0 as a parameter in their drift-diffusion simulations. While I still believe that it would be desirable to verify the experimentally observed effects with drift-diffusion simulations with a constant energy gap instead of a variable V_0 (such as the model put forward by Blakesley and Neher, ref. 41), using an experimentally determined V_0 would be acceptable in my opinion.*

Response: Thank you for the comment. The Dieter Neher work was in fact a key basis for our understanding of the drift-diffusion-generation model. The natural (and good) starting point is to assume constant energy gap for pinning, as in that work and the rest of the literature. However, we are here evaluating the impact of soft pinning that emerges as a consequence of driving work function of the electrode to extreme values, causing Fermi level to detach from the usual assumption of fixed level pinning and creep up the DOS tail of the semiconductor. As a consequence, the pinning “gap” narrows slightly. This obliges us to consider a variable V_0 which we then peg to the experimentally measured low-temperature diode built-in potential. The change with work function is small, increasing from 1.13 V to 1.25 V, for work function increasing from 5.2 eV to 5.9 eV, but this is significant for solar cells...

2. *In Supplementary Fig. 12, the effect of the contact work function on thin films is not visible, while the mobility in thin films should be more sensitive to the charge density at the contact compared to thick films. This set of data would rather point to a transition from a slightly non-Ohmic regime to an Ohmic regime, as opposed to the “post-Ohmic” regime as the authors put forward. In SCLC measurements it is hard to distinguish whether an increased current is due to a higher injection rate or an increased mobility. Usually thickness dependence gives a good clue, but in this case the evidence would be in favor of an injection limitation due to slightly non-Ohmic contacts. This would of course also have an effect on the mobility due to the reduced charge density, but this would not be the dominant effect in the magnitude of the current.” and “Another implication of a density-dependent mobility would be a non-constant mobility over the measured current density range, as is assumed in the fitting.*

Response: Thank you for this query. We started out on the same assumption as referee

that film thickness dependence of JV characteristics would provide a sensitive test for Ohmic contact. Then we realized that the assumption of constant mobility function is often violated. Decreasing film thickness of polymers below certain critical value often causes mobility to decrease, although particularly pronounced for PBDTPD films, which also show a change in absorption spectrum—features sharpen, suggesting better “order” which may ironically kill mobility because of emergence of traps in-between the ordered regions (Suppl Fig 5).

Therefore, we rely on the Mott-Gurney index $m = \frac{d \log J}{d \log (V - V_*)}$, which simulation takes a value close to 2.0 whenever the injecting contact is not limiting, i.e. it can maintain carrier density at the contact higher than some threshold, and carriers in the bulk exhibit an average mobility that does not vary over the tested voltage range. The range is typically only a few volts, corresponding to up to two orders of magnitude of variation in current density, about one order of magnitude of variation in current density. Such analyses show the contacts are indeed Ohmic, $m \approx 2$, even at 50-nm thickness (Supp Fig 11). Thus the apparent loss in mobility at thicknesses smaller than 80 nm is real. The low mobility suggests presence of deep traps. Depending on shape and intensity of these traps, they can stop Fermi level creeping up, and therefore suppress the contact effect. This is not investigated further, because it looks complicated while the interesting physics occurs outside this thickness regime (fortunately).

The reviewer made another thought-provoking comment: “Another implication of a density-dependent mobility would be a non-constant mobility over the measured current density range”. We have now incorporated full carrier-density-dependent mobility in our drift-diffusion-generation simulation. Guided by the numerical results in Pasveer PRL 2005, we found a convenient empirical function to represent the density dependence of the mobility: $\mu_p = \mu_{p,0} \left(1 + \alpha \left(\frac{n_p}{N} \right)^{0.50} \right)$, which for $\alpha = 21$ is consistent with $\sigma/kT = 3$, where n_p is carrier density, and N is the transport site density ($2.0 \times 10^{20} \text{ cm}^{-3}$). This allows us to show: (i) Indeed the JV characteristics over the measured current—voltage range exhibits $m = 2.0$ for Ohmic contacts but with apparent mobility that increases with carrier density at the contact. [In contrast, once the contact becomes non-Ohmic, the JV evolves towards linear as the contact resistance starts to dominate.] (ii) The simulations with the above ‘universal’ function reproduce the solar cell characteristics very well across the entire work function range, with no ‘free’ fitting parameter (Fig 5). In the previous version of our manuscript, we had used empirically determined “constant” apparent mobility for each work function. Comparing the two results, we now know that the apparent mobility is about 75% of the density-dependent mobility at the contact for the film thickness in these cells. As a consequence, we can now firmly conclude that the mobility effect we have observed in both hole-only diodes and solar cells stem from a single origin in carrier-density mobility dependence, due to

the high carrier density that can now be accumulated at the contact that has not previously been observed. We thank the reviewer for “pushing” us to clarify this.

3. *These points of criticism are not to say that the results are invalid, and the authors have now convincingly demonstrated that for these face-stacked polymers extreme contact work functions can be used to extract more performance from such solar cells. I would, however, avoid using terms like "post-Ohmic" in a revised version of the manuscript.*

Response: In view that the devices have now been demonstrated to be operating in the Ohmic regime from work function of about 5.2 eV onwards, it seems justified to give a name to the phenomena that occur in this regime. We think the referee is right that it should not be called “post-Ohmic” because the phenomenon still lies inside the Ohmic regime, but it is “post-Ohmic transition” in the sense that it occurs well after the Ohmic transition is crossed. We have amended the manuscript accordingly. So in summary, in general one has to reach the Ohmic transition, beyond Fermi-level pinning, to minimize contact resistance and get best performance for organic solar cells. This is general. However, under certain conditions that result in soft Fermi level pinning, e.g. for the face-stacked polymer PBDTPD here, going well beyond the Ohmic transition is advantageous, as it opens a slightly larger V_{oc} due to opening of the Fermi-level pinning gap. In addition, one can generally raise carrier mobility by squeezing Fermi level up the DOS to get a larger FF in the process, without running immediately into a voltage loss due to the counteracting electrostatic band bending.

4. *How would a more conventional MoO₃/Ag electrode compare in terms of performance in an inverted device architecture? MoO₃ has also been reported to have an extremely high work function.*

Response: We have now fabricated an “conventional” inverted solar cell with the structure of ITO/ZnO/PBDTPD:PCBM/MoO₃/Ag (Suppl Fig 9). We also compared this to a corresponding solar cell ITO/ZnO/PBDTPD:PCBM/HCL-5.75/Ag, where HCL-5.75 is the spin-on hole-doped polymer described in the manuscript with work function of 5.75 eV. The MoO₃/Ag cell gives $V_{oc} = 0.93$ V and FF = 0.60, which are both lower than the HCL-5.75/Ag cell, which gives $V_{oc} = 0.97$ V and FF = 0.66, and lower than all cells with Ohmic hole collection contact in the main report.

However the MoO₃/Ag cell gives a higher $J_{sc} = 9.7$ mA cm⁻², compared to HCL-5.75/Ag cell, $J_{sc} = 7.65$ mA cm⁻². This is due to the deposition of the polymer layer causing a surface loss and rearrangement of the PCBM.

The ‘surprising’ observation is although MoO₃ gives even higher work function than the spin-on hole-doped polymer, it gives poorer V_{oc} and FF characteristics. We believe this is due to deeper Fermi-level pinning, and the formation of a surface chemistry layer that is

associated with a larger contact resistance, which appears to be consistent with the findings of Blom and co-workers [Kotadiya, Nat Mater (2018)]. However we cannot say much more, because we don't really know what is happening at the semiconductor/MoO₃ contact.

5. *What assumptions/values are used for the bimolecular recombination rate in the simulations? I could not find these details (maybe I have overlooked), but this is important information.*

Response: Fixed. The geminate recombination rate constant used was $1 \times 10^5 \text{ s}^{-1}$, bimolecular recombination rate used was $2 \times 10^{-11} \text{ cm}^3 \text{ s}^{-1}$, now included in the caption of Table 1.

Reviewer #4 (Remarks to the Author):

We thank this review for his/her critical comments on the discussion of Fermi level pinning in our manuscript. We have now revised the manuscript accordingly.

1. *In the entire introduction the authors fail to clearly explain state-of-the-art knowledge of FL pinning. While in the "old" literature FL pinning was often alleged to be abrupt (seen from the z-like curves plotted with an abrupt change in slope), it was later on clearly stated (also in papers cited by the authors) that any density of states that is not abrupt leads to a further movement of FL with electrode work function change. This is irrespective of shape of the majority DOS (e.g., Gaussian), and particularly governed by sample-dependent gap state density (which can be intrinsic or extrinsic, see work by the Ueno group), it's shape and actual density. This, for instance, was shown explicitly for an inorganic semiconductor quite recently (doi.org/10.1103/PhysRevMaterials.3.074601; and I just realize that even the term "soft FL pinning" was used there). So there is no new physics here, but the authors demonstrate the impact of "soft" FL pinning on V_{bi} and V_{oc} of OPV. This has not been shown yet.*

Response: The Ueno group has done beautiful work on gap states, but the phenomenon we are discussing here is not the consequence of such gap states but the tail of the main transport states. Most of the community does not pay so much attention to the phenomenon of soft Fermi level pinning. For example, in the review by Yang, Bussolotti, Kera and Ueno, J Phys D (2017), which we have now included in the set of 'classic reviews' in the introduction of our paper, the schematic illustration of Fermi level position in the HOMO–LUMO gap (Figure 4) illustrates only the situation for abrupt pinning. When the disorder width becomes large, the transition from $S=0$ to $S=1$ becomes rounded, which we call soft Fermi level pinning. This is in fact seen in the calculations by Ueno's group in Figure 7 of the paper, when the disorder width σ becomes larger than 0.2 eV, as also shown by many other authors, but so far no body has paid much attention to the consequences of this rounding for devices because it was not thought to be important. Typical semiconductors show disorder width less than 0.15 eV, in best semiconductors, less than 0.1 eV, so such a phenomenon was thought to be theoretically possible but practically irrelevant. The review article therefore also did not discussed much further about this phenomenon, but directs attention to fixed Fermi level pinning at 2σ as a general phenomenon. In fact this report appears to be the first to give this phenomenon a name, soft Fermi level pinning.

The article doi.org/10.1103/PhysRevMaterials.3.074601 cited by the reviewer does mention the term "soft Fermi level pinning", but for a very different phenomenon. There it refers to the situation when the gap states at the surface of ZnO is not large enough to rigidly pin the Fermi level, which can therefore change by large amount, depending on

the surface adsorbate, resulting in correspondingly large change in band bending in the sub-surface region.

What we are discussing here is not gap states, but the tail of the main states which generally determines Fermi level because of its high density.

New science:

- (1) Although the possible theoretical existence of soft Fermi level pinning is trivial, this has not been discussed in the literature, and never been experimentally demonstrated previously. We show in this work for the first time an important situation where this arises merely from DOS broadening of the frontier monolayer of the semiconductor, for example by face stacking, which is a common feature of 'modern' donor and acceptor polymers for PAL. Bulk broadening, which ruins the semiconductor, is not required. From materials point-of-view, the new appreciation here is that face-stacked semiconductors can have the DOS sufficiently broadened (few tenths of an eV) at the interface in devices to yield soft Fermi level pinning, even while the bulk DOS is rather narrow (0.075 eV)! We have revised the manuscript to improve clarity. There is no other report like this.
- (2) The effects of such a soft FL pinning on device physics are not trivial, and has never been elucidated in the literature. They lead to a better Voc, because the shift can be larger than the counteracting electrostatic band bending. And unexpectedly, it can populate the DOS tail states sufficiently to raise carrier mobility. From device physics point-of-view, what is new is the realization that the density-dependence of carrier mobility can be exploited through contact accumulation to raise the mobility in devices simply by using electrodes with extreme work function. [MoO₃ does not work because of its strong trapping of the accumulated carriers.] This has never been demonstrated previously.

We have also cited a paper by Kera and Ueno now to explain why we cannot rely on UPS to give the correct DOS width, due to vertical ionization and other effects, but instead use transport modelling.

2. *It is established by numerous publications that (low) doping of an organic semiconductor increases carrier mobility due to filling of shallow trap states. This is, from a fundamental point of view, generally valid and fully independent of how doping is done, whether it is with dopants or by inducing FL pinning (the present case). So there is no new physics here, but the authors demonstrate the impact of this contact-doping on the performance of OPV.*

Response: Yes, the referee is right. All these phenomena fundamentally come from filling up of the DOS tail, whether by chemical doping, gate field, or current density. This is however the first report that suggests contact carrier accumulation also can fill up tail

states sufficiently to induce higher mobility in polymer organic semiconductor thin films to make a difference (improvement) to organic solar cells. This is such simple approach! But this simple physics does not appear to have previously been appreciated, perhaps because systematic studies with well-defined ultrahigh work function electrodes were not available. Maybe it can be considered to be of some general interest.

REVIEWERS' COMMENTS:

Reviewer #2 (Remarks to the Author):

The authors have answered my remaining questions and, in my opinion, the manuscript can now be published.